# Anisotropic Low Cycle Behavior of the Extruded 7075 Al Alloy

**DOI:** 10.3390/ma14164506

**Published:** 2021-08-11

**Authors:** Jin Ma, Qiang Wang, Yongbiao Yang, Fulai Yang, Beibei Dong, Xin Che, Hui Cao, Tingyan Zhang, Zhimin Zhang

**Affiliations:** College of Materials Science and Engineering, North University of China, 3 Xueyuan Road, Taiyuan 030051, China; nucmajin@163.com (J.M.); nucyangfulai@163.com (F.Y.); dongbb1111@163.com (B.D.); CXIN19931029@163.com (X.C.); 18434365466@163.com (H.C.); nuczty@163.com (T.Z.); ZZMNUC@163.com (Z.Z.)

**Keywords:** 7075 Al alloy, anisotropy, texture, low cycle fatigue, fatigue model

## Abstract

The quasi-static and low cycle fatigue tests of extruded 7075 Al alloy (Φ200 mm) were investigated in three directions: the extrusion direction (ED), the radial direction (RD), and 45° with ED (45°). Grain morphology analysis, texture measurement, and fatigue fracture characterization were conducted to discuss the relationship between microstructure and mechanical properties. The experimental results showed that the ED specimen had higher ultimate tensile strength (UTS) and low cycle fatigue (LCF) properties, which were mainly attributed to the following three causes. First, the grain boundaries (GBs) had an obvious effect on the crack growth. The number of GBs in the three directions was different due to the shape of the grains elongated along the ED. Second, the sharp <111> texture and the small Schmidt factor along the ED explained the higher ultimate tensile strength (UTS) of the ED specimens. Third, fatigue fracture observation showed that the ED specimen had a narrow fatigue striation spacing, which indicated that the plastic deformation of the ED specimen was the smallest in each cycle. In addition, two fatigue prediction models were established to predict the LCF life of extruded 7075 Al alloy, based on the life response behavior of the three directions under different strains.

## 1. Introduction

Owing to the low density and high specific strength of aluminum (Al) alloy, it has been extensively applied to reduce the weight and improve the transportation capacity and mobility in many fields. Many structural components are in a state of tension–compression cyclic loading for a long time during the service process, which could easily cause fatigue failure and fracture of materials [1]. In order to offer the higher-level of safety to these industries during service, it is particularly important to detect the fatigue performance of these kinds of alloys [2,3,4]. In addition, due to the forming methods of Al alloy components, such as rolling, extrusion, etc., their mechanical properties have great anisotropy, which limits the application of Al alloy in most industries.

Low cycle fatigue experiments of Al alloy under strain control have been investigated for many years. A considerable number of research has been carried out to investigate the effect of microstructure and metallurgical state on age hardening Al alloy during cyclic loading [5,6,7,8,9,10,11]. Sadeler et al. [9] mentioned the effect of the relevant heat treatment process on the fatigue properties of AA2014 Al alloy. The fatigue strength of the alloy treated by solution and aging treatment was 43% higher than that of the as cast alloy at 10^7^ cycles. The effects of grain morphology, texture, and second phase precipitation on the fatigue properties of the material have also been discussed. Some papers also researched the effect of pretreatment on the fatigue behavior of Al alloy. The influence of surface roughness and residual compressive stress on the fatigue properties of the specimens by surface shot peening was investigated [12]. The experimental results showed that the fatigue life of the specimens increased significantly after 180 s of surface shot peening. Ludian et al. [7] demonstrated that shot peening could improve the high-cycle fatigue performance of AA2024 Al alloy. Sreenivasan et al. [13] discussed the influence of ratcheting strain on the low-cycle fatigue behavior of 7075-T6 Al alloy specimens. The results showed that the fatigue life of the specimens decreased significantly after ratcheting. However, these previous studies were focused on Al alloy plates or small-diameter bars, and there is still a lack of research on large-diameter Al alloy bars.

At the same time, the research on the anisotropy of mechanical properties has also been increasing. In such papers, the causes of anisotropy and the methods to weaken the anisotropy were studied. H. Takahashi et al. [14] compared the evolution of plastic anisotropy of Al alloy plates by evaluating the flow stress of the tensile test in various directions and showed that unstable lattice rotation would cause the anisotropy to become sharper. Zhang et al. [15] investigated the difference in mechanical properties of 2024 plate welded joints; Fourmeau et al. [16] analyzed the plastic anisotropy of AA7075-T651 Al alloy in different directions; El-Aty et al. [17] revealed the deformation behavior, strengthening mechanism and anisotropy of Al-Li alloy. These tests showed that the anisotropy of the mechanical properties was related to the shape and orientation of the grains. However, there is still a lack of research on the anisotropic low cycle behavior of large-diameter Al alloy bars.

This paper aimed to study the quasi-static and strain-controlled low-cycle fatigue properties of Φ200 mm extruded Al alloy (7075-T6) in three directions: the extrusion direction (ED), the radial direction (RD), and 45° with ED (45°). The relationship between properties and microstructure was established. The stress–strain hysteresis loops and stress amplitude-number of the cycles response is rigorously discussed in three directions. Finally, two fatigue prediction models, Smith–Watson–Topper (SWT) and Jahed–Varvani (JV) [18], are given to predict fatigue life.

## 2. Materials and Methods

In this paper, the material was an extruded 7075 Al alloy bar (Table 1) with a diameter of 200 mm and a height of 4000 mm. In order to meet the actual conditions of Al alloy bar in production and reduce the influence of natural ageing on the test result, the experimented alloy was conducted by T6 (480 °C × 6 h + 130 °C × 24 h) treatment to ensure the stability of the properties of the specimens.

Quasi-static and fatigue smooth dog-bone specimens were designed as in Figure 1a,b. For research on the anisotropy of the Al alloy bar under different loading directions, the specimens were examined along three directions, namely ED (extrusion direction), RD (radial direction), and 45° (45° to the ED). Considering that the microstructure of the large diameter bar varies greatly in different positions along the RD, the gauge sections of all specimens were situated at the same diameter. The extraction locations of specimens in the extruded 7075Al alloy are illustrated in Figure 1c. Before the fatigue test, the defects on the surface due to the machining could be eliminated by hand grinding parallel to the load axis with gradually finer SiC paper.

The microstructure of specimens was characterized by optical microscopy (OM, DM2500M, Leica Microsystems, Wetzlar, Germany) and scanning electron microscopy (SEM, SU5000, Hitachi, Tokyo, Japan). The specimen was ground with gradually finer SiC paper with grit No. up to #5000 and polished with 1 μm diamond paste. The etchant was acidic solution: 1 mL HF, 1.5 mL HCl, 2.5 mL HNO_3,_ and 95 mL H_2_O. The specimens analyzed by electron backscattered diffraction (EBSD) required electropolishing after grinding with an electrolyte (90% C_2_H_5_OH + 10% HClO_4_) at 20 V for ~20 s at −18 °C. The accelerating voltage of all EBSD tests was 20kV and the scanning step was 1 μm. The original data of the EBSD tests could be processed by the orientation imaging microscopy (OIM, EDAX Inc., Mahwah, NJ, USA). The compositions of the phases and compounds in the alloy were analyzed using energy dispersive spectroscopy (EDS, EDAX Inc., Mahwah, NJ, USA).

Quasi static and low cycle fatigue tests at the same ambient temperature were carried out on Instron 3382 and Instron 8801 servo hydraulic testing machines (Instron Inc., Canton, MA, USA) with 100 kN load capacity, respectively. The strain rate was maintained at 5.0 × 10^−3^ s^−1^, and the strain was measured by the corresponding extensometer which had a gauge length of 12.5 mm and a range of ±10%. Strain control was used in the low cycle fatigue test and the waveform was sinusoidal, while the strain ratio was −1. Five different strain amplitudes were adopted, 0.5%, 0.6%, 0.7%, 0.8%, and 0.9%, respectively. In order to ensure the same strain rate in all experiments, the frequency was decreased with the increase of strain amplitude, from 0.25 Hz to 0.139 Hz. A few double-sided tapes were pasted at the contact point between the tool edge of the extensometer and the gauge distance of the specimen to avoid early fatigue crack [19]. Fatigue life was set at 50% load reduction or final fracture. For each strain condition, at least three experiments should be done to ensure the accuracy of the results. After failure, both the tensile and fatigue fracture morphology of the specimens was analyzed by SEM.

## 3. Results and Discussion

### 3.1. Microstructure

#### 3.1.1. Metallographic Structure

The OM of the heat-treated specimens on the ED-RD and RD-RD of extruded 7075 Al alloy is given in Figure 2. In the ED-RD plane, most grains elongate along ED and are fibrous with an average thickness of 50 µm. The grains on the RD-RD plane are equiaxed, and the average grain size is 60 μm. Many large-sized second phases are distributed in the grain boundaries, and a few small-sized second phases are distributed in the grains’ interior.

Figure 3a,b is OM and SEM of the second phase, respectively. The composition of these second phases was identified by an energy dispersive spectrometer (EDS) and the results are shown in Figure 3c. The ratio of Al to Cu is close to 2:1, so it can be considered as Al_2_Cu. The second phases of Al_2_Cu can hinder dislocation movement, which is an important reason for the increase in the static mechanical properties [20,21].

#### 3.1.2. EBSD Analysis

Figure 4a,c shows the inverse pole figure (IPF) of RD-RD and RD-ED, respectively. Black lines represent the high angle grain boundaries (HAGB, >15°), and white lines indicate the low angle grain boundaries (LAGB, 2°~15°). The HAGB has an obvious effect on the fatigue crack growth, which has been widely recognized by the academic community [22]. The grain boundary resistance of the crack propagation along the ED is larger than that along the RD. Therefore, the difference of HAGB density in the three directions is an important reason for the anisotropy of the mechanical properties. Figure 4b,d shows the (111) pole figure (PF) of RD-RD and ED-RD specimens respectively, which indicates that there is a strong texture in the aged extruded 7075 Al alloy bar, and the (111) crystal plane is perpendicular to the ED. Assuming that the texture type of the whole material is {111} <110> texture, the reason for the anisotropy can be well explained.

Figure 5 is the Schmid factor of sliding system {111} <110> for different specimens along the loading direction. The Schmid factor reflects the difficulty of activating a specific sliding system under loading. The small Schmid factor corresponds to hard orientation for this sliding system which is difficult to activate [23]. The more difficult the slip system is to activate, the higher is the strength of the specimen. The average Schmid factors of ED, 45° and RD are 0.39, 0.47, and 0.45, respectively. Therefore, the strength of the ED specimen is the highest.

### 3.2. Quasi-Static Test

The quasi-static engineering stress–strain curves in the three directions and corresponding elongation (EL), yield strength (YS), ultimate tensile strength (UTS) and Young’s modulus are shown in Figure 6 and Table 2. The UTS of the ED is higher than that of the RD and the 45°, and the EL of the 45° is higher than that of the ED and RD. The quasi-static mechanical properties of specimens in the three directions show strong anisotropy.

Figure 7 shows the macroscopic fracture morphologies of the specimens in the three directions. The fracture sections of the ED, the 45°, and the RD are 45° to the loading direction, so the fracture mechanisms of the three directions are the same, and belong to ductile fracture. However, the fracture morphology can be seen to show great differences in the three directions. Comparing the fracture of specimens in the three directions, it can be observed that the fracture surface of the 45° specimen shows obvious necking phenomenon, so the plasticity of the 45° specimen is the best and the EL is the highest. However, the fracture surface of the ED and RD specimens shows no necking phenomenon, so the plasticity is relatively poor compared to the 45°specimen. However, compared with the fracture of the ED and RD specimens, the fracture of the RD specimen is straighter, so it can be considered that the plasticity of the RD specimen is worse than that of the ED specimen.

Figure 8 is an enlarged view of the tensile fracture in the three directions. For all specimens, there are a lot of dimples and cleavage planes in the fracture surface, which can be considered as a mixed mode of intergranular fracture and transgranular fracture. Due to the difference of grain orientation, the grain boundary resistance of the specimens in the three directions is different, which shows the anisotropy of the mechanical properties [24]. In Figure 8, it can be clearly observed that cracks are generated at the grain boundary. The grain boundary morphology of the ED and 45° specimens is similar to the OM of the RD-RD plane, so it can be proved that the fracture of the 45° specimens is perpendicular to the ED. The mechanical properties are related to the microstructure in the three directions. When the stress direction is along the long axis of the grain, the crack is perpendicular to the grain boundary. In order to have a crack through the whole section, the crack must pass through multiple grain boundaries. So, the UTS of the ED is the highest. The cleavage plane of the RD specimens is larger than that of the 45° and ED specimen, corresponding to a worse EL.

### 3.3. LCF Behavior

The summary of fatigue life and deformation parameters in the three directions is shown in Table 3. Generally speaking, the fatigue life of alloy will decrease with the increase of strain amplitude. The ED specimens have better fatigue resistance compared with the RD and 45° specimens. Furthermore, the anisotropy of the fatigue life is to be further analyzed.

#### 3.3.1. Hysteresis Loops

Based on the results of quasi-static behavior, the appropriate strain is selected for the low cycle fatigue test. The selected strain is controlled in the elastic deformation zone of the 7075 Al alloy. When the strain amplitude is less than 0.4%, the fatigue life is too high, which does not belong to the scope of low cycle fatigue in this paper. When the strain amplitude is greater than 0.9%, the specimen will fracture within a few weeks, which is meaningless for this study. Figure 9 shows the hysteresis loops of the extruded 7075 Al alloy under strain-control at different strain amplitudes and direction, and the area of hysteresis loops indicates the level of cycle plastic deformation of the specimens [25]. The black dash-dot lines, the red full lines, and the blue dashed lines represent the first cycle, the second cycle, and the half-life cycle, respectively. From Figure 9, all hysteresis loops are symmetrical under different strain amplitude, which indicates that 7075 Al alloy extruded in different directions has a similar deformation behavior during the tension and compression stage. However, the area of the half-life cycle loops is smaller than those of the second cycle, which indicates that the plastic strain of the half life cycle is lower.

With the increase of strain amplitude, the hysteresis loop area of the specimens in the three directions increases, which indicates that the plastic deformation also increases. When the strain amplitude is 0.5%, the hysteresis loop curve is approximately linear, which indicates that the proportion limit of Al alloy is relatively high and the specimens are still in the elastic stage while plastic deformation is almost absent under cycle deformation. The Young’s modulus of tension and compression can be obtained from the results of the first cycle fatigue test in the three directions, as shown in Figure 9a,d,h. The Young’s modulus in tension is greater than that in compression, so the average stress is negative in the whole process of the fatigue test. On increasing strain amplitude to 0.7%, the hysteresis loop becomes plump and smooth, and the greater the plumpness of the hysteresis loop, the better the plasticity of the material, which indicates the materials enter the elastic-plastic stage. When the strain amplitude reaches 0.9%, the specimen will fracture within several hundred cycles, and the cyclic deformation will be dominated by plastic strain.

Moreover, this is essential to compare the hysteresis loops response of the ED, the 45°, and the RD specimens at the same strain amplitude. When the strain amplitude is 0.5%, it can be seen from the data in Table 3 that the stress amplitude of the ED specimens is greater than that of the 45° and RD specimens, indicating that the extruded 7075 Al alloy has a higher Young’s modulus along the ED. With the increase of strain amplitude, the stress amplitude in the ED specimen is much larger than that in the 45° and RD specimens, which indicates that extruded 7075 Al alloy has a higher YS along the ED. The hysteretic area of the first cycle and half life cycle of the ED specimen is obviously less than 45° and the RD specimens are under the same strain amplitude when the strain amplitudes are 0.7% and 0.9%, which indicates that the plastic strain of each cycle of the ED specimens is much less than the 45° and RD specimens. This phenomenon can be directly observed in the fracture morphology at the later stage.

#### 3.3.2. Cyclic Stress Response

Figure 10 shows the relationship between the number of cycles and the stress of the specimens in the three directions under the same strain. The stress amplitude of the specimen increases and the fatigue life decreases with increasing total stain amplitude. However, under the same strain amplitude, the stress amplitude of the ED specimen is bigger than the 45° and RD specimens. With the increase of strain amplitude, the difference of stress amplitude increases gradually. Furthermore, the higher YS is, the higher the threshold of the stress amplitude will be, and the lower the plastic deformation will be. It can be proved that increasing the YS can improve the low-cycle fatigue life of materials.

When the stain amplitude is 0.5%, the cyclic hardening characteristics of the ED specimens are obvious, and continue until the failure of the specimen. There is no cyclic softening phenomenon. The stress amplitudes of the ED, 45° and RD specimens are almost the same at the beginning. With the progress of the experiment, the stress amplitudes of ED are nearly 20 MPa larger than those of the 45° and RD. There is no obvious phenomenon of cyclic hardening or cyclic softening in the 45° and RD specimens, and the cyclic stress amplitude remains constant.

When the stain amplitude is 0.7%, early cyclic softening and cyclic hardening behaviors can be observed in the ED, 45° and RD specimens. The cyclic softening occurs in the early stage of the experiment. When it reaches about 50 cycles, cyclic softening ends and cyclic hardening starts synchronously until the specimens are broken. At the same time, it can be observed that the stress amplitude of the RD specimens is bigger than that of the 45° specimens and lower than that of the ED specimens. The evolution law of cyclic stress amplitude of the 45° and RD specimens is the same, and the difference of stress amplitude is almost the same in the whole process. However, the evolution law of the cyclic amplitude of the ED specimens is different from the other two directions. At the beginning of the cycle, the stress amplitude of the ED specimens is 60 MPa bigger than that of the RD specimens. However, with the progress of the experiment, the difference of the stress amplitude gradually decreases to 40 MPa.

When the strain amplitude is 0.9%, the variation of the stress amplitude is similar to that of the strain amplitude at 0.7%. However, the end of the cyclic softening behavior is advanced, and the difference of cyclic stress amplitude is larger. It has been previously reported that Al 7075-T6 with low plastic strain amplitude (less than 0.6%) undergoes cyclic hardening and saturation before fracture, while for plastic strain with an amplitude greater than 0.6%, cyclic softening may occur [26]. The behaviors observed in this study are the same as those reported above.

According to the above results, the cyclic stress amplitude along the ED of extruded 7075 Al alloy is always higher than that of the 45° and RD. This is due to the fibrous grain morphology and the sharp texture along the ED, as described in the 3.1.3. The difference of cyclic stress evolution between the ED and the 45° and the RD is due to the difference of plastic deformation under the same strain amplitude. The amount of plastic deformation in the ED direction is much smaller than that in the other two directions, which can be directly observed from the data listed in Table 3. In addition, at low strain amplitude, the evolution of the cyclic behavior is mainly determined by the dislocation structure. The hardening behavior in Figure 10 is mainly attributed to the deformation strengthening mechanism. Dislocation movement and multiplication are caused by deformation, which means that with the stress to overcome dislocation mutual movement increases and leads to cyclic hardening [12,27]. After T6, there are fine precipitates in the extruded 7075 Al alloy. Higher strain amplitude may lead to the dissolution of precipitates, resulting in cyclic softening [26,28].

#### 3.3.3. Fatigue Fracture Morphologies

Three typical areas of fatigue fracture, including fatigue crack initiation (FCI) area, fatigue crack growth (FCG) area, and final fracture (FF) area, can be clearly seen in Figure 11. The small area surrounded by the red dotted line is the FCI area, which is close to the surface of the specimen. The possible location of the FCI is usually a defect on the surface of the specimen or the location of the second phase. In the process of deformation, the deformation is discontinuous, which leads to the initiation of fatigue crack. The black arrow shows the direction of fatigue crack growth. The FCG area is a fan-shaped area from the FCI area. The yellow dotted line is used to separate the crack propagation area and the FF area. The size of the FCG area often represents the fatigue life of the specimen [18,29]. The FCG area of the 45° and RD specimens is similar, which is larger than that of the ED specimen. However, according to Table 3, the fatigue life of the ED specimen is much higher than that of the 45° and RD specimens, which is mainly due to the different plastic deformation in each cycle. It is shown in the fatigue fracture that the size of the striation spacing is different.

Figure 12a,d,g shows enlarged views of the FCI area of specimens in the ED, 45° and RD, respectively. FCI is the core of the streamline emitted from the center which in the three directions is near the surface of the specimens. The second phase (Al_2_Cu) and the surface quality of the specimens are the important factors causing the cracks. Before the fatigue test, the surfaces of all specimens are polished to eliminate the influence of surface quality. The second phase can be observed during the crack initiation. Due to the different lattice parameters between the second phase and the matrix, continuous deformation cannot be realized during the deformation process, which may produce stress concentration in the second phase, leading to the initiation of fatigue crack.

Figure 12b,e,h shows enlarged views of the FCG areas of specimens in the ED, 45°, and RD, respectively. It can be clearly seen that the fatigue striations (FS) are perpendicular to the direction of crack propagation. The tension and compression load makes the fracture engage in a process of blunting and sharping, which leads to the generation of fatigue striations. The fatigue striation spacing represents the crack growth of each cycle [30].

Figure 12c,f,i shows enlarged views of the FF areas of specimens in the ED, the 45° and the RD, respectively. There are dimples, tearing ridges and secondary cracks on the fracture surface, which are similar to the fracture of quasi-static tension.

### 3.4. Fatigue Life Prediction

As mentioned above, the fatigue properties of extruded 7075 Al alloy bar have strong anisotropy in the three directions, and the fatigue life is quite different. Therefore, it is of great significance to establish a model to predict the fatigue life in the three directions. In this paper, the two fatigue life prediction models, Smith–Watson–Topper (SWT) and Jahed–Varvani (JV), were used. Through the existing data in getting the model parameters the accuracy of the two models was evaluated.

#### 3.4.1. Smith–Watson-Topper (SWT)

The total strain in the loading process is divided into elastic strain and plastic strain. The relationship between elastic strain amplitude, plastic strain amplitude, and fatigue life (2N_f_) can be expressed by the Basquin equation (Equation (1)) and the Coffin–Manson equation (Equation (2)), respectively.
(1)$$\frac{\Delta \varepsilon_{e}}{2}=\frac{\sigma_{f}^{\prime }}{E}{\left(2N_{f}\right)}^{b}$$
(2)$$\frac{\Delta \varepsilon_{p}}{2}=\varepsilon_{f}^{\prime }{\left(2N_{f}\right)}^{c}$$
where $\sigma_{f}^{\prime },b,\varepsilon_{f}^{\prime }$ and c are the fatigue strength coefficient, fatigue strength index, fatigue ductility coefficient and fatigue ductility index respectively. According to the previous static performance calculation, E is about 74 GPa. The formulation of total strain is as follows:(3)$$\frac{\Delta \varepsilon }{2}=\frac{\Delta \varepsilon_{e}}{2}+\frac{\Delta \varepsilon_{p}}{2}=\frac{\sigma_{f}^{\prime }}{E}{\left(2N_{f}\right)}^{b}+\varepsilon_{f}^{\prime }{\left(2N_{f}\right)}^{c}$$

SWT is an improvement of the Manson coffin relation, which is based on the product of the tensile strain amplitude by the peak normal stress. The corresponding formula of the SWT model is as follows:(4)$$SWT=\sigma_{n,max}\frac{\Delta \varepsilon }{2}=\frac{\sigma_{f}^{\prime }{}^{2}}{E}{\left(2N_{f}\right)}^{2b}+\sigma_{f}^{\prime }\varepsilon_{f}^{\prime }{\left(2N_{f}\right)}^{b+c}$$

The relationship between the elastic and plastic strain amplitude of the specimens relative to fatigue life (2*N_f_*) and the SWT prediction model are listed in Figure 13.

#### 3.4.2. Jahed–Varvani (JV)

JV is a fatigue life prediction model based on energy damage criterion. The total strain energy density is divided into two parts: positive elastic strain energy density $\left(\Delta E_{e}^{+}\right)$ and plastic strain energy density $\left(\Delta E_{p}\right)$. The corresponding formula of the JV model is as follows:(5)$$\Delta E=\Delta E_{e}^{+}+\Delta E_{p}=\frac{\sigma_{max}^{2}}{2E}+\oint \sigma d\varepsilon =E_{e}^{\prime }{\left(2N_{f}\right)}^{B}+E_{f}^{\prime }{\left(2N_{f}\right)}^{C}$$
where $E_{e}^{\prime },B,E_{f}^{\prime }\;and\;C$ are the tensile peak stress, fatigue strength coefficient, fatigue strength index, fatigue ductility coefficient and fatigue ductility index based on the energy respectively. The functional relationship between the elastic and plastic strain energy density of the specimens relative to fatigue life (2*N_f_*) and JV prediction model are listed in Figure 14.

Figure 13d and Figure 14d show the comparison between the fatigue life predicted by the SWT and JV models and the experimental life of the three direction specimens, respectively. The solid diagonal line indicates a perfect match, while the blue and green dotted line indicates a deviation coefficient of 1.5 and 2 between the predicted life and the experimental life. Most data points gather at the boundary with a coefficient of 1.5, which shows that both models are suitable for the life prediction and analysis of extruded 7075 Al alloy.

## 4. Conclusions

Quasi-static tensile and strain-controlled LCF properties of extruded 7075 Al alloy in the ED, 45° and RD were investigated to evaluate the fatigue anisotropy. The following conclusions can be drawn from this paper.

The extruded 7075 Al alloy exhibited sharp <111> texture along the ED, which leads to a small Schmid factor in the loading direction and increases the difficulty of activating the sliding system. The strength of the ED specimen was better than that of the 45° and RD specimens.In each cycle, the specimen of ED has the smallest plastic deformation, so it has the longest fatigue life. The fatigue life of the 45° specimen was second only to the ED, which was mainly due to the fact that the better plasticity in the 45° could accommodate bigger plastic deformation before failure.The position of crack initiation was on the surface or on the second particles beneath the surface for all specimens which indicated that the precipitates were the important factors affecting fatigue crack initiation.The SWT critical plane fatigue life model based on the strain and JV fatigue life model based on energy damage criterion showed good fitting accuracy in extruded 7075 Al alloy, which can be used as a reliable low cycle fatigue life prediction model.

## Figures and Tables

**Figure 1 materials-14-04506-f001:**
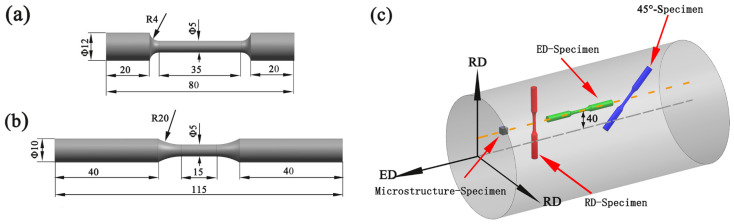
Specimen designs for (**a**) tensile tests, (**b**) fatigue tests; (**c**) The extraction locations of specimens in the extruded 7075 Al alloy (Dimensions are in “mm”).

**Figure 2 materials-14-04506-f002:**
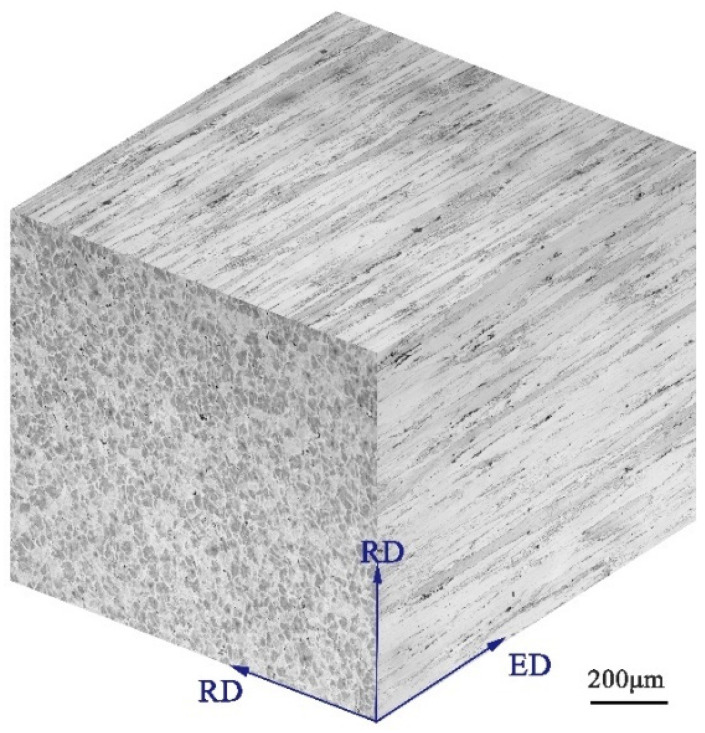
Three-dimensional optical micrograph of the extruded 7075 Al alloy.

**Figure 3 materials-14-04506-f003:**
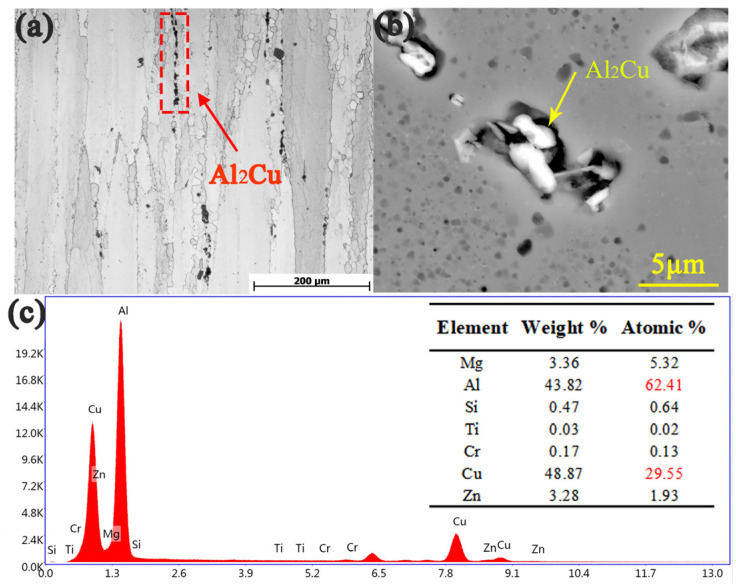
(**a**) OM and (**b**) SEM micrographs on the RD-ED plane of the extruded 7075 Al alloy, (**c**) EDS spot scan results of particles.

**Figure 4 materials-14-04506-f004:**
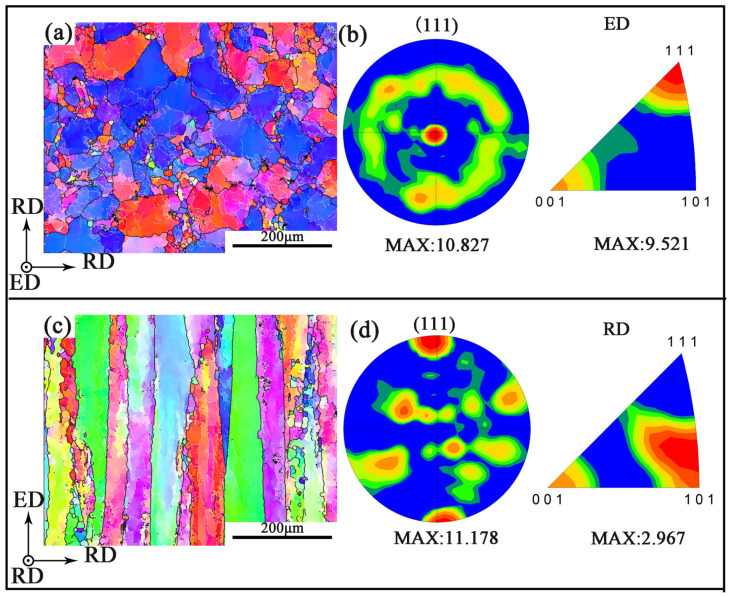
The inverse pole figure (IPF) and the corresponding (111) pole figure (PF) of the specimens. (**a**,**b**) RD-RD. (**c**,**d**) ED-RD.

**Figure 5 materials-14-04506-f005:**
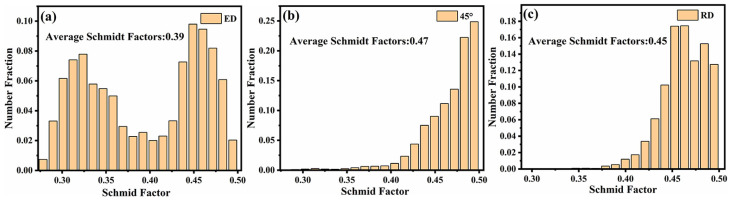
Schmid factor of sliding system {111} <110> for different specimens: (**a**) ED (**b**) 45°, and (**c**) RD.

**Figure 6 materials-14-04506-f006:**
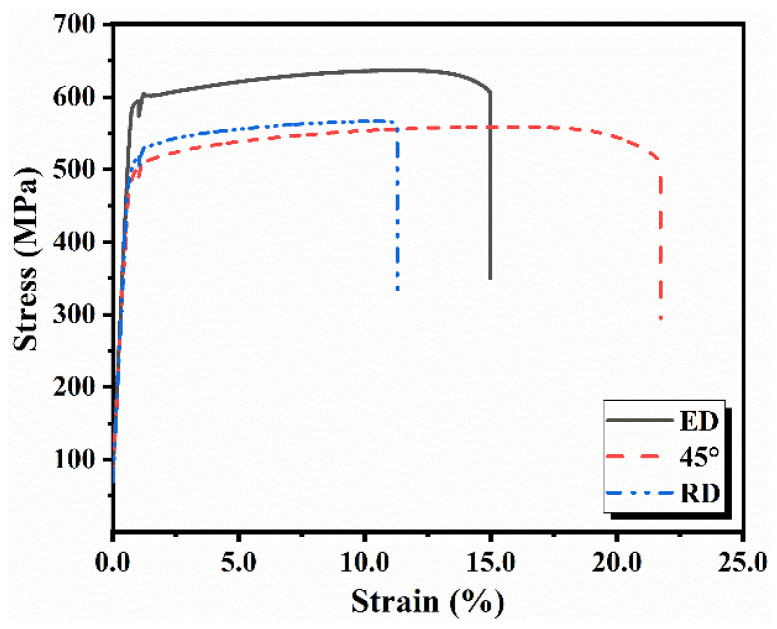
Stress–strain curves of the ED, 45°, RD specimens under quasi-static tension.

**Figure 7 materials-14-04506-f007:**
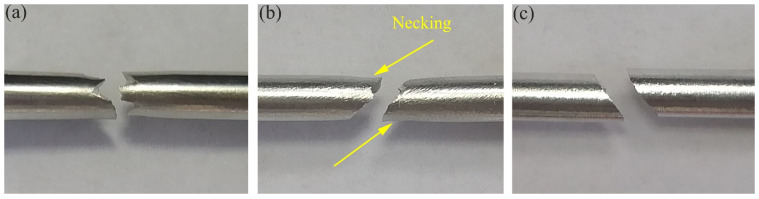
Macroscopic fracture morphologies of extruded 7075 Al alloy in different directions: (**a**) ED, (**b**) 45°, (**c**) RD.

**Figure 8 materials-14-04506-f008:**
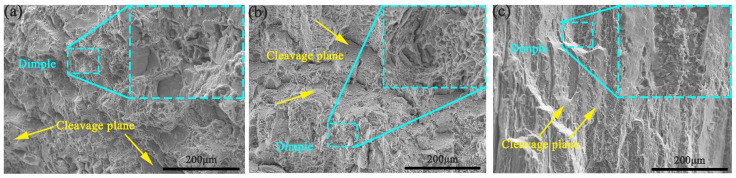
Tensile fracture morphologies of extruded 7075 Al alloy in different directions: (**a**) ED, (**b**) 45°, (**c**) RD.

**Figure 9 materials-14-04506-f009:**
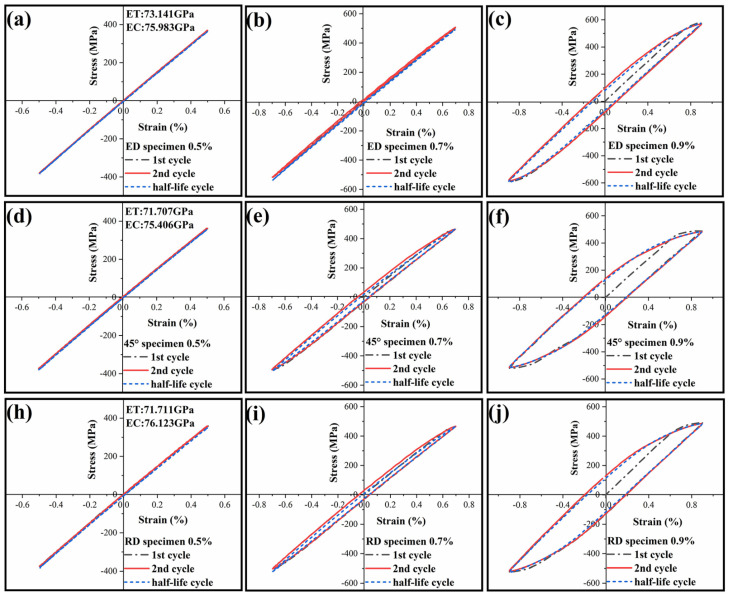
The evolutionary change of stress-strain hysteresis loops for the ED (**a**–**c**),45° (**d**–**f**) and RD (**h**–**j**) specimens at different total strain amplitudes of (**a**,**d**,**h**) 0.5%, (**b**,**e**,**i**) 0.7%, and (**c**,**f**,**j**) 0.9%.

**Figure 10 materials-14-04506-f010:**
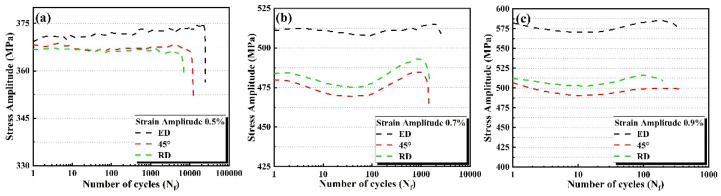
The stress amplitude vs. number of cycles for (**a**) 0.5%, (**b**) 0.7 and (**c**) 0.9% specimens under different strain amplitudes.

**Figure 11 materials-14-04506-f011:**
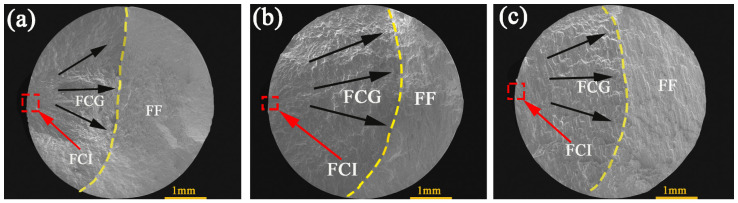
SEM images of fatigue fracture surfaces of the 7075 Al alloy tested at a total strain amplitude of 0.5% for (**a**) ED, (**b**) 45°, and (**c**) RD specimens.

**Figure 12 materials-14-04506-f012:**
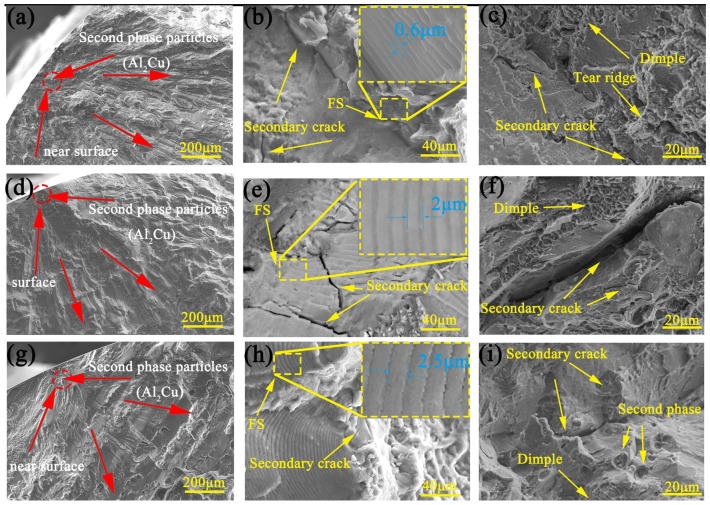
SEM images of (**a**,**d**,**g**) FCI, (**b**,**e**,**h**) FCG and (**c**,**f**,**i**) FF region at a total strain amplitude of 0.5% for ED (**a**–**c**), 45° (**d**–**f**) and RD (**g**–**i**) specimens.

**Figure 13 materials-14-04506-f013:**
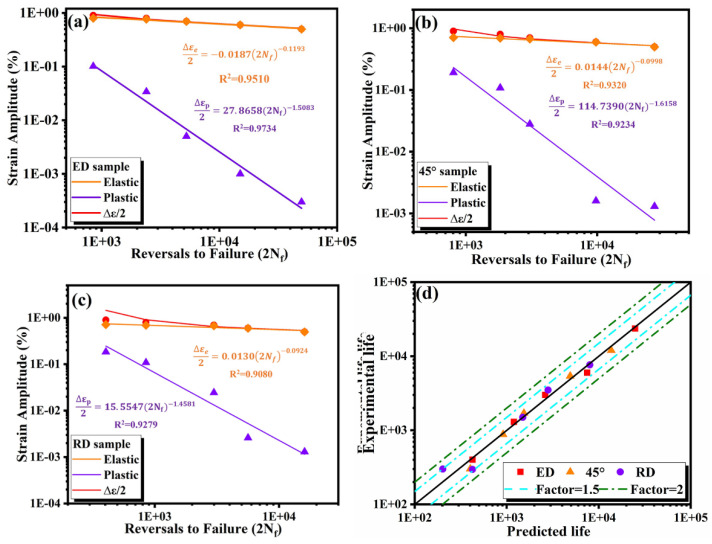
SWT prediction model for (**a**) ED (**b**) 45°(**c**)RD (**d**) SWT predicted vs. experimental life.

**Figure 14 materials-14-04506-f014:**
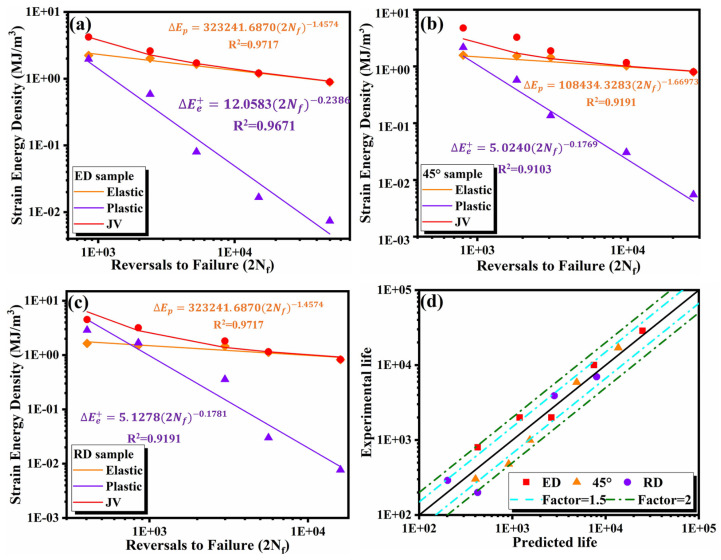
JV prediction model for (**a**) ED (**b**) 45° (**c**) RD (**d**) JV predicted vs. experimental life.

**Table 1 materials-14-04506-t001:** Chemical composition of the 7075 Al alloy (wt%).

Element	Zn	Mg	Cu	Mn	Fe	Si	Cr	Ti	Al
Content	5.97	2.07	1.58	0.27	0.32	0.11	0.14	0.028	Bal

**Table 2 materials-14-04506-t002:** Quasi-static tensile properties of extruded 7075 Al alloy. (The numbers in the parentheses are standard deviations).

Specimen Type	YS (MPa)	UTS (MPa)	EL (%)
ED	592(1)	636(2)	15(1)
45°	491(0)	559(1)	22(0)
RD	508(1)	566(1)	11(0)

**Table 3 materials-14-04506-t003:** Fatigue tests summary for the half-life cycle. (The numbers in the parentheses are standard deviations).

Specimen	Δε/2 (%)	$\sigma_{max}\;(\mathrm{MPa})$	$\sigma_{min}\;(\mathrm{MPa})$	$\Delta \varepsilon_{e}/2\;(\%)$	$\Delta \varepsilon_{p}/2\;(\%)$	$\Delta E_{e}^{+}\;(\mathrm{MJ}/m^{3})$	$\Delta E_{p}\;(\mathrm{MJ}/m^{3})$	N_f_
ED	0.50	362.22	−388.06	0.50	0.01	0.89	0.01	24,894
		(1.04)	(1.30)	(0)		(0.21)	(0)	(2054)
	0.60	420.52	−468.40	0.60	0.01	1.19	0.02	7508
		(0.69)	(0.52)	(0)		(0.2)	(0)	(1045)
	0.70	492.60	−537.38	0.69	0.01	1.64	0.08	2625
		(0.43)	(0.32)	(0.01)		(0.42)	(0.01)	(257)
	0.80	547.42	−590.04	0.77	0.03	2.02	0.58	1199
		(0.96)	(1.02)	(0.01)		(0.91)	(0.02)	(100)
	0.90	575.27	−595.15	0.80	0.10	2.24	1.97	425
		(1.35)	(1.22)	(0.03)		(0.94)	(0.07)	(30)
45°	0.50	343.55	−391.25	0.49	0.01	0.80	0.01	13,673
		(0.74)	(0.51)	(0)		(0.11)	(0)	(1420)
	0.60	411.16	−455.01	0.59	0.01	1.14	0.03	4896
		(0.30)	(1.20)	(0)		(0.09)	(0)	(783)
	0.70	462.42	−506.50	0.67	0.03	1.44	0.44	1536
		(0.67)	(1.21)	(0)		(0.20)	(0.04)	(300)
	0.80	475.62	−514.49	0.69	0.10	1.53	1.74	913
		(0.74)	(0.55)	(0.01)		(0.32)	(0.09)	(103)
	0.90	483.72	−515.28	0.71	0.19	1.58	3.17	402
		(1.20)	(1.2546)	(0.03)		(0.27)	(0.20)	(60)
RD	0.50	349.40	−382.83	0.49	0.01	0.82	0.01	8012
		(0.30)	(0.22)	(0)		(0.12)	(0)	(1350)
	0.60	407.63	−470.17	0.59	0.01	1.12	0.03	2815
		(0.62)	(0.77)	(0)		(0.25)	(0)	(709)
	0.70	465.23	−520.44	0.68	0.02	1.46	0.36	1495
		(1.38)	(0.85)	(0.01)		(0.27)	(0.01)	(330)
	0.80	473.01	−512.55	0.69	0.11	1.51	1.69	425
		(1.20)	(1.21)	(0.01)		(0.22)	(0.17)	(90)
	0.90	493.53	−539.37	0.72	0.18	1.65	2.88	202
		(1.62)	(1.32)	(0.02)		(0.14)	(0.21)	(20)

## Data Availability

Date is contained within the article.

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
