# Peer review of "Anisotropic Low Cycle Behavior of the Extruded 7075 Al Alloy"

_materials, 2021, doi:10.3390/ma14164506_

Round 1

Reviewer 1 Report

The paper clearly reports an experimental study, which was carefully run. The results are  interesting, although the analysis remains  superficial. In particular,  is a pitty that short cracks growth and  their interactions with the grain boundaries were not monitored on the outer surface. Some improvements are needed for publication.

On page 8, line 218, the authors write that the stress-strain loops are symmetrical in tension and compression. But on page 9, line 230, they write that the average stress is negative (which is not visible on figure 9). This is contradictory.

To what mechanism do the authors attribute the secondary hardening observed during some tests (Fig. 10b and c). Was such a secondary hardening already reported for this material in the literature? Is it a real hardening or just the consequence of crack initiation outside the gage length of the extensometer an a compensation of the control software?

The approach presented in paragraph 3.4.1 has nothing to do with Smith-Watson and Topper's criterion (based on the product of the tensile strain amplitude  by the peak normal stress). It is Manson & Coffin's model.

In Table 3, the stresses and strains are reported with an unreasonable number of digits, considering the accuracy on their measurement.

The scales along the y axis of figure 10 (especially 10a) are not well chosen. Reduce the range. Furthermore, the black and blue colors are too similar.

During crack growth under fixed strain range (and decreasing stress range), the striation spacing has no reason to remain constant.  It is thus not meaningful to report a single spacing for a given specimen, and to use it for a comparison  between different specimens. The paragraph starting on line 327 page 11, and the conclusion page 14, line 389 should be removed.

The English  should be substantially improved

Reviewer 2 Report

The paper by Wang Qiang, Yang Yongbiao et al. reports an experimental investigation on 7075 extruded Al alloy with the aim to evaluate the fatigue anisotropy. The paper is clearly written, with good quality data and an exhaustive discussion. I have only minor remarks:

- tables 2 and 3 do not report the uncertainties associated to each quantity. The authors should show (or at least give an estimation of) the error bars either directly obtained from the experiments or by propagation of the errors;

- symbols in line 350 do not appear in previous eqs. 1 and 2;

- figure 15 (line 369) is missing in the text;

- there are some typos: line 42: 107? Maybe 10 million?   Line 188: ’intergranular fracture is repeated twice.
